# Impact of Donor and Host Age on Systemic Cell Therapy to Treat Age-Related Macular Degeneration

**DOI:** 10.3390/cells14171360

**Published:** 2025-09-01

**Authors:** Carolina Francelin, Xiaoping Qi, Juliana Godoy, Brenton T. Bicknell, Ram Prasad, Maria B. Grant, Michael E. Boulton

**Affiliations:** 1Department of Ophthalmology and Visual Sciences, University of Alabama at Birmingham, Birmingham, AL 35294, USA; xqi@uabmc.edu (X.Q.); juliana.godoy@einstein.br (J.G.); brentonb@uab.edu (B.T.B.); ramprasad@uabmc.edu (R.P.); mariagrant@uabmc.edu (M.B.G.); 2Hospital Israelita Albert Einstein, Sao Paulo 04829-310, SP, Brazil

**Keywords:** hematopoietic stem cells, bone marrow-derived cells, progenitors, retinal degeneration, age-related macular degeneration, aging, gene therapy, retinal pigment epithelium, RPE65

## Abstract

**Purpose:** We previously reported that the systemic administration of preprogrammed mouse hematopoietic bone marrow-derived progenitor cells (HSPCs) improved visual function and restored a functional retinal pigment epithelial (RPE) layer. Here, we investigated the potential impact of donor vs. host age on systemic cellular therapy in a murine model of retinal degeneration. **Methods:** HSPCs from young (8 weeks) and old (15 months) mice were programmed ex vivo with a lentiviral vector expressing the RPE65 gene (LV-RPE65) and systemically administering into young or old SOD2 KD mice. Visual loss and pathological changes were evaluated by electroretinogram (ERG), optical coherence tomography (OCT), histology, and immunohistochemistry. **Results:** Old donor HSPCs administered to old manganese superoxide dismutase (SOD2) knockdown (KD) recipient mice offered the least benefit. This was exemplified by the reduced recruitment and incorporation of LV-RPE65 HSPC into the RPE layer, as well as decreased improvement in visual function, retinal thinning, and limited reduction in oxidative damage and microglial activation. LV-RPE65 HSPC from young mice incorporated into the RPE layer of old SOD2 KD mice, though to a lesser extent than young cells administered to young hosts, offered some level of protection. By contrast, LV-RPE65 HSPCs from old mice, located to the subretinal space of young host mice, reduced visual loss, although some retinal pathology was observed. **Conclusions:** The administration of LV-RPE65 HSPC from old donors to old SOD2 KD mice offered the least improvement. **Translational Relevance:** Our findings highlight how both donor and recipient age impact the success of HSPC-based retinal therapy and using cells from aged donors for AMD treatment may have some limitations.

## 1. Introduction

We previously reported that the systemic administration of mouse hematopoietic bone marrow-derived progenitor cells (HSPCs), defined by the surface markers cKit^+^Sca-1^+^, when infected ex vivo with a lentiviral vector expressing the RPE-specific gene RPE65 (LV-RPE65), demonstrate rapid recruitment to the injured retina and improve visual function by restoring a functional RPE layer in both acute and chronic murine models of retinal degeneration [1,2,3]. These studies were performed using donor HSPC from young animals injecting into young mice (approx. 8- to 12-weeks-old) undergoing retinal degeneration models. Numerous studies have reported that the donor and host ages of stem/progenitor cells influence the capacity of the cells to mediate repair and their utility as cellular therapy. Typically, beneficial outcomes are reduced with the increasing age of either the donor or the host [4,5,6,7,8,9,10,11,12]. This is particularly relevant since therapies are designed to target dry age-related macular degeneration (AMD), a disease that impacts over 170 million globally and with an incidence that increases after 60 years of age [13,14,15]. Normal age-related changes in the retina include Bruch’s membrane thickening, RPE changes, and photoreceptor loss which, in dry AMD, include the following: drusenoid deposits, the further thickening of Bruch’s membrane, lipofuscin accumulation, inflammation, and eventually, RPE cell loss, resulting in geographic atrophy [16,17]. Previously, we reported that LV-RPE65 HSPCs were most effective in preventing visual loss in mice when administered in early retinal disease compared to during late retinal degeneration [2]. Aging also impacts the stem/progenitor’s self-renewal, differentiation, and recruitment capabilities, as well as increasing senescence [8,15,16,17]. Bone marrow-derived cells have been extensively used in allogeneic hematopoietic cell transplants, and there is clear evidence that younger donor age is associated with improved transplant-related outcomes and decreased relapse risk [4,8,9,18]. In addition, HSPCs from old mice are less able to engraft when compared to cells from young mice [19]. Furthermore, particularly relevant to our studies utilizing intravenous administration of HSPCs, there is an age-related decline in blood flow [20,21,22], which has the potential to reduce the rate and number of LV-RPE65 HSPCs recruited to the retina following systemic administration. Given that AMD occurs in the elderly and given the need to harvest allogeneic peripheral blood cells or bone marrow from elderly donors, we investigated the potential impact of donor vs. host age on systemic cellular therapy in a murine SOD2 KD model of retinal degeneration. Our studies identified that LV-RPE65 HSPCs from young mice were reparative in both young and old SOD2 KD mice with an AMD-like phenotype, while LV-RPE65 HSPCs from old mice were reparative in young SOD2 KD mice but repair was minimal in old recipient mice.

## 2. Methods and Materials

### 2.1. Animals

Eight-week-old and fifteen-month-old C57BL/6J and homozygous green fluorescent protein (GFP) (C57BL/6-Tg [UBC-GFP]) female mice were purchased from Jackson Laboratory and maintained at the University of Alabama in Birmingham Animal Resources Program. All mouse studies were performed under the approval of the Institutional Animal Care and Use Committee (IACUC, protocol # 20912) from the University of Alabama at Birmingham in accordance with the National Institutes of Health and the Association for Research in Vision and Ophthalmology guidelines for the use of animals in ophthalmic and vision research.

### 2.2. SOD2 KD Mouse Model

The right eye of female C57BL/6J mice was injected sub-retinally with 0.6 µL of 2.5 × 10^10^ particles/mL of recombinant AAV1 constructs based on the pTR-UF2 vector expressing SOD2-specific hammerhead ribozyme, Rz432, driven by an RPE-specific VMD2 promoter (AAV1-ribozyme (Rz)-SOD2) to drive ribozyme gene expression efficiently in the RPE layer as previously described [2,23,24]. Control injections of rAAV-inactive ribozyme (AAV1-Rz-inactive) were performed in the right eye of the control cohort.

### 2.3. Preparation of RPE65-Programmed HSPC

HSPCs were programmed ex vivo by inserting a stable RPE65 transgene using a lentiviral vector, as we have previously described [2,25]. In brief, HSPC were isolated aseptically from bone marrow from the tibiae and femurs of GFP^+^ mice, according to our standard protocols, and the linage (Lin^−^) population was enriched using EasySep Mouse Hematopoietic Progenitor Cell Enrichment Kit (StemCell, Vancouver, BC, Canada, #19856C) [3]. The enriched cell suspension was then stained with PE-conjugated rat anti-mouse Ly-6A/E (BD Bioscience, Franklin Lakes, NJ, USA, #561076) and the positive cells were sorted as GFP^+^ Stem Cell Antigen-1^+^(Sca1^+^) using the Becton Dickinson and Company Fluorescence-Activated Cell Sorting (BD FACS) Melody in the Flow Cytometry and Single Cell Core Facility at the University of Alabama at Birmingham. Lentiviral constructs expressing RPE65 (TYF-RPE65) and LacZ (TYF-EZ-LacZ) were prepared as previously described [2,3]. The sorted Lin^−^GFP^+^Sca1^+^ cells were suspended in Dulbecco’s Modified Eagle Medium (DMEM) (high glucose) and polybrene (8 μg/mL) plus 10% fetal bovine serum at a final cell concentration of 5 × 10^4^ cells/mL. For the infection, the HSPCs were split into three 1 × 10^5^ cell aliquots. Two aliquots were infected with 5 μL of recombinant lentivirus expressing TYF-EZ-LacZ or 2 μL of lentivirus expressing TYF-RPE65, respectively, at a multiplicity of infection of ∼50. The third aliquot received 5 μL of phosphate-buffered saline (PBS) (pH 7.4) as vehicle control. After washing and resuspension, cells were injected into mice as described below. We did not observe any difference in infection efficiency between old and young HSPC.

### 2.4. Systemic Administration of RPE65-Programmed HSPC

At 1 month following SOD2 KD, mice received a systemic injection of 5 × 10^4^ Lin^−^Sca1^+^GFP^+^ cells (either naive HSPC, LV-LacZ HSPC, or LV-RPE65 HSPC) in 50 μL of sterile saline systemically administered into the tail vein (a minimum of *n* = 5 per group) as we previously described [2,25]. Negative controls included the injection of the inactive ribozyme (Rz-Inactive) and untreated age-matched wild-type C57BL/6J mice.

### 2.5. Electroretinography (ERG)

Mouse ERGs were recorded at 1 and 3 months following cell injection as we have previously described [2,24]. Mice were dark-adapted overnight, and full-field ERGs were recorded with a visual electrodiagnostic system (Universal Testing and Analysis System (UTAS)-E 2000; LKC Technologies, Gaithersburg, MD, USA). In brief, gold wire loop electrodes were placed on each cornea, and a reference electrode was placed subcutaneously between the eyes and another electrode at the tail. The scotopic rod recordings were carried out when the light stimuli were presented at intensities of 0.025, 0.25, and 2.5 log cd-s/m^2^ at intervals of 10, 20, and 30 s, respectively. A total of 10 responses were recorded and averaged at each light intensity. The photopic cone recordings were performed after light-adaption at 100 mcd-s/m^2^ for 7 min. An average of 50 responses was recorded from each flash intensity of 0, 5, 10, and 25 log cd-s/m^2^ with constant 100 mcd-s/m^2^ background light and averaged for each intensity. Final results were expressed as follows: a-wave—by the measurement from the baseline to the peak in the cornea-negative direction; b-wave—by the measurement from the cornea negative peak to the major cornea-positive peak. The means of the maxima for the a and b wave responses of each animal/group were plotted and compared between all groups.

### 2.6. Optical Coherence Tomography (OCT)

OCT analysis was performed at 1 and 3 months following cell injection, as previously described [2,24]. In brief, mice were anesthetized by ketamine (72 mg/kg)/xylazine (4 mg/kg), and their eyes were dilated with a solution of 1% atropine plus 2.5% phenylephrine hydrochloride (Alcon, Fort Worth, TX, USA), followed by one drop of 2.5% hydroxypropyl methylcellulose (Gonak^®^, Akorn, Lake Forest, IL, USA) for lubrication. Retinal scans were performed using a Bioptigen OCT system (Envisu R-class, Leica Microsystems, Buffalo Grove, IL, USA) and three lateral images (nasal, central, and temporal) were taken, starting at 0.23 mm above the meridian crossing through the center of the optic nerve (ON), at the ON meridian, and 0.23 mm below the ON meridian. A corresponding box centered on the ON with twenty-five measurement points separated by 0.24 mm from each other were taken. Total neural retinal thickness was measured from the vitreous face of the ganglion cell layer to the apical face of the RPE. The retinal thickness average/scan/mouse was plotted and compared between all groups.

### 2.7. Histopathology

At 3 months following cell injection, the mice were euthanized and their eyes were enucleated and fixed in freshly prepared 4% paraformaldehyde at 4 °C for 24 h. Following 3 washes in PBS, the eyes were sent to the Histology Core Facility of the University of Alabama at Birmingham for routine paraffin wax processing and sectioning. Five micron sections were cut and stained with hematoxylin and eosin. The histopathological analysis was performed in sections at the 3-month time point only. Stained sections were examined by light microscopy (Zeiss Axio Imager 2, Oberkochen, Germany) and representative images taken approximately 200 µm from the optic nerve.

### 2.8. Immunohistochemistry

Immunohistochemistry was conducted on wax-embedded sections of mouse eyes as we previously described [2,24]. In brief, the sections were deparaffinized and incubated with 1X Rodent deblocker (Biocare Medical, LLC, Pacheco, CA, USA) for 45 min at 120 °C, followed by five minutes on the bench to cool down. Sections were then permeabilized with 0.1% Triton X-100 for 10 min, washed in PBS, and blocked with 5% normal goat serum in PBS plus 3% BSA for 1 h at room temperature. Blocking solution was removed without washing and the sections received the primary antibody made with the blocking solution. For double staining, rabbit-anti mouse RPE65, diluted to 1:400 (Abcam, Cambridge, United Kingdom #231782), combined with chicken-anti mouse GFP, diluted to 1:500 (Abcam, Cambridge, United Kingdom #13970), were added to the sections overnight at 4 °C. Sections were additionally stained with the following antibodies: anti-4 hydroxynonenal antibody (4HNE, catalog # STA-035, Cell Biolabs, San Diego, CA, USA), polyclonal rabbit antibody, and ionized calcium-binding adapter molecule 1 (Iba1) (#019-19741, Wako Chemicals, Richmond, VA, USA). The sections were then washed with PBS plus 0.25% Tween and incubated, as appropriate, with either Alexa Fluor 555 goat anti-rabbit IgG (Invitrogen, Waltham, MA, USA #A-21428), Alexa Fluor 488 goat anti-chicken (Invitrogen, Waltham, MA, USA #A-11039), or Alexa Fluor 488 goat anti-mouse (Invitrogen, Waltham, MA, USA #A-11001), diluted to 1:1000, and Hoechst solution of 20 mM (Pierce, Rockford, IL, USA #62249) in blocking solution for 1 h at room temperature. After extensive washing with PBS, cells were mounted with either ProLong^TM^ Glass antifade Mountant (Invitrogen, #P36980) or VectaShield 4′,6-diamidino-2-phenylindole (DAPI) (Vector Laboratories, Newark, CA, USA) for nuclear staining. Negative control samples were processed with the omission of the primary antibody. The sections were examined using a Zeiss fluorescence microscope or a Nikon AX-R Laser Confocal microscope (Nikon, Tokyo, Japan) with identical settings for intensity, gain, etc. Representative images were taken approximately 200 µm from the optic nerve. The quantification of immunofluorescence staining for the measurement of anti-4-HNE was performed by a masked observer in paraffin-embedded tissue sections with 10 representative images from each sample at 20X magnification then imported into ImageJ (developed by Wayne Rasband, National Institutes of Health, Bethesda, MD, USA; available at https://imagej.net).

### 2.9. Statistical Analysis

All experiments were repeated up to three times. The results are all expressed as means ± SEM. Samples were evaluated by applying Grubb’s test to determine any outliers, followed by an unpaired two-tailed Student’s *t*-test or ANOVA; ordinary one-way ANOVA post hoc correction were applied, as appropriate, to determine statistical significance of the data. All analyses were performed using GraphPad Prism 9 (GraphPad Software, La Jolla, CA, USA), with *p* values of ≤0.05 considered statistically significant.

## 3. Results

### 3.1. Improvement in Visual Function

For this study, we utilized both young and old mice as recipients of HSPC injections and utilized donor HSPCs isolated from either young or old mice. HSPCs were either infected with LV-Lac Z as control or LV-RPE65 as the experimental virus. Previously, we reported that in the SOD2 KD model, which involves the subretinal injection of AAV1-Rz-SOD2, a significant reduction in the scotopic a- and b-waves was observed compared to normal untreated controls and compared to mice receiving the inactive ribozyme [2]. This reduction is observed at both the 1- and 3-month time points. In the current study, we observed a reduction in the photopic b-wave in the eyes of young SOD2 KD mice but we did not see any reduction in photopic b-wave in the eyes of old SOD2 KD mice at 1 month; however, by 3 months post AAV1-Rz-SOD2 injection, there was a reduction in the photopic b-wave (Figure 1). A small decrease in scotopic visual function was observed in animals receiving the inactive ribozyme but this was significantly less than that for SOD2 KD animals and was consistent with our previous studies in young mice [2]. The administration of LV-RPE65 HSPCs from old mice into young SOD2 KD mice demonstrated a significant improvement in the scotopic a- and b-wave, but not the photopic b-wave, compared to control (Figure 1A). LV-RPE65 HSPCs from young mice administered to old SOD2 KD mice demonstrated a significant improvement in both scotopic a- and b-waves and also photopic b-wave compared to SOD2 KD mice not administered HSPCs (Figure 1B). LV-RPE65 HSPCs from old mice administered into old SOD2 KD mice resulted in a significant improvement in scotopic a-wave at 1 month only and b-wave at both 1 and 3 months compared to untreated SOD2 knockdown mice (Figure 1C). No significant improvement was observed in the photopic b-wave at either 1 or 3 months following HSPC administration.

### 3.2. Improvement in Retinal Structure and Morphology

In agreement with our previous study [24], OCT analysis demonstrated that compared to controls and untreated animals, both old and young SOD2 KD mice presented the significant thinning of the retina (*p* ≤ 0.05), at both the 1 and 3 month time points examined (Figure 2A). Pathological changes included outer retinal changes and the loss of photoreceptors and RPE. The changes became progressively more severe with increasing time following SOD2 KD. Retinal thickness from untreated or mice receiving inactive ribozyme did not demonstrate structural changes during the period of analysis (Figure 2B,D,F). At 3 months following the administration of LV-RPE65 HSPCs from old mice to young SOD2 KD mice, significantly reduced retinal thinning was observed compared to LV-LacZ HSPCs or untreated SOD2 KD mice and retinal structure appeared normal (Figure 2A,B). LV-RPE65 HSPCs from young mice administered to old SOD2 KD mice prevented the retinal thinning and structural changes at both 1 and 3 months post cell administration when compared to the untreated group (Figure 2C,D). This is similar to what we previously reported for HSPCs from young mice administered to young SOD2 KD mice [2]. In the case of LV-RPE65 HSPCs from old mice to old SOD2 KD mice, retinal thinning was reduced compared to untreated SOD2 KD mice and retinal structure appeared normal (Figure 2C). However, old SOD2 KD mice receiving LV-LacZ HSPCs from old donors showed improved retinal thickness compared to SOD2 KD mice without cell injections, but there was additional benefit in maintaining retinal thickness in the SOD2 KD mice injected with LV-RPE65 HSPC (Figure 2E,F).

Histological analysis demonstrated that both young and old SOD2 KD mice 4 months after SOD2 KD (equivalent age to mice 3 months after HSPC injection) demonstrated retinal thinning, photoreceptor loss, and RPE depigmentation/atrophy compared to untreated control or mice receiving inactive Rz-SOD2 as we previously reported (Figure 3) [2,23]. As previously reported, SOD2 KD mice receiving LV-LacZ HSPCs showed a similar morphology to that of untreated SOD2 KD mice. No change in neural retina and RPE morphology was observed in untreated and Rz-inactive mice groups (Figure 3A,B). Neither old SOD2 KD mice receiving LV-RPE65 HSPCs from young mice nor young SOD2 KD mice receiving LV-RPE65 HSPCs from old mice presented with any major change in neural retina and RPE layer morphology, although in the case of young mice receiving old LV-RPE65 HPSCs, some cells were observed in the subretinal space overlying the RPE (Figure 3). However, while LV-RPE65 HSPCs from old mice administered to old SOD2 KD mice did show less pathology compared to Rz-SOD2-only mice, there remained some neural retina thinning and RPE monolayer changes at 3 months (Figure 3B,C).

### 3.3. Localization of HSPCs to the RPE Layer in SOD2 KD Mice

We next performed immunostaining for (1) an RPE cell phenotype using anti-RPE65 and (2) HSPC^GFP+^ using an anti-GFP antibody. RPE65 staining confirmed the loss of RPE cells from the monolayer in both young and old SOD2 KD mice eyes after 3 months of reprogrammed HSPC injections (Figure 4). As we previously reported [2], 3 months after the administration of LV-RPE65 HSPCs from young mice to young SOD2 KD mice, a relatively normal and continuous RPE layer occurred compared to untreated or LV-LacZ HSPC administered SOD2 KD mice (Figure 4A). GFP-positive staining, indicative of integrated HSPC, was observed at the RPE layer in LV-RPE65 HSPC SOD2 KD retinas and GFP-positive cells colocalized with RPE65 staining (Figure 4A). However, 3 months after the administration of LV-RPE65 HSPCs from old mice to young SOD2 KD mice, a relatively normal and continuous RPE layer occurred that stained RPE65 but not GFP (Figure 4B). GFP-positive cells were observed in the subretinal space but had not integrated into the RPE layer. By contrast, 3 months after the administration of LV-RPE65 HSPCs from young mice to old SOD2 KD mice, GFP-positive cells colocalized with RPE65 staining and were integrated into the RPE monolayer with no cells observed in the subretinal space (Figure 4C). However, there appeared to be fewer LV-RPE65 HSPCs compared to young HSPCs administered to young mice. Although the RPE layer appeared relatively normal in old SOD2 KD mice receiving old LV-RPE65 HSPCs, no GFP-positive cells were detected (Figure 4D). No GFP-positive cells were observed in the RPE layer or the subretinal space of SOD2 KD mice receiving LV-LacZ HSPC (Figure 4E).

### 3.4. Reduction in Oxidative Damage and Inflammation

Consistent with our previous study [24], the immunohistochemical detection of 4-hydroxynonenal (4-HNE), a marker of oxidative stress, showed a very low level of 4-HNE immunostaining (red) in control retinas of both young and old mice while SOD2 KD retinas demonstrated a significant increase (*p* < 0.01) in 4-HNE staining that was not reduced by treatment with LV-LacZ HSPCs (Figure 5A–D). 4-HNE staining was dramatically reduced in the young and old retinas of SOD2 KD mice after receiving old or young LV-RPE65 HSPCs, respectively (Figure 5B,D), and was statistically significant at *p* < 0.05. While a reduction in 4-HNE staining was observed in the retina of old mice receiving LV-RPE65 HSPC from old mice this was not statistically significant compared with SOD2 KD mice or SOD2 + LV-LacZ HSPC (Figure 5E,F). In addition, the immunostaining of retinal sections detected an increased number of Iba1-positive (Iba1^+^) microglia after SOD2 KD both in old and young mice (Figure 6A,B). These Iba1^+^ cells exhibited the morphological characteristics of activated microglia with large cell bodies plus ramification and were located both in the inner retina and subretinal space (Figure 6). SOD2 KD eyes receiving either old or young LV-RPE65 HSPCs showed fewer and smaller Iba1^+^ cells, which was not observed in eyes treated with either old or young LV-LacZ-HSPCs (Figure 6).

## 4. Discussion

In keeping with our previous studies, administering HSPCs from young donor animals to young host SOD KD mice [1,2,3] resulted in the reprogrammed HSPCs being recruited to the retina and resulted in reduce visual loss and retinal degeneration in a mouse model with an AMD-like phenotype. However, neither old donor HSPCs nor old hosts performed as well as young donors and/or hosts. In a previously published paper, we demonstrated that photopic a-wave ERG responses are improved in young SOD2 KD mice receiving LV-RPE65 HSPCs from young donors [2]. This was not the case for old host SOD2 KD mice receiving LV-RPE65 HSPCs from young donors. The reason for this is unclear, especially since LV-RPE65 HSPCs were recruited to the RPE in the young donor HSPCs to old SOD2 KD hosts and retinal morphology was significantly improved. Second, the recruitment and integration of RPE65 HSPCs appear to be age-dependent: (1) old LV-RPE65 HSPCs administered to young mice appeared to be localized to the subretinal space above the RPE; (2) no GFP-positive cells were observed integrated into the RPE or in the subretinal space, raising the possibility that these cells had initially been recruited, as there was some limited improvement in visual function and morphology, but did not survive the full 3 months; and (3) the observation that the photopic b-wave in old SOD2 KD mice treated with RPE65-HSPCs is significantly higher than that of the untreated control may suggest the improved function of cone-driven ON bipolar cells 1 month after treatment. Third, LV-RPE65 HSPC from old mice administered to old SOD2 KD mice did not significantly reduce oxidative damage, as was observed in the other groups. These observations are perhaps not surprising since it is well recognized that increased host and donor age can have a significant negative impact on cellular therapy approaches [4,8,9,15,16,17,18]. Aging affects the capacity for the self-renewal and differentiation of stem cells, decreasing the potential for regeneration and resulting in a loss of optimal function. HSPCs exhibit promising anti-aging properties by targeting the underlying mechanisms of retinal aging, including the modulation of chronic inflammation and oxidative stress as shown here but also cellular senescence [26]. While not tested here, HSPCs may exhibit their regenerative potential by their immunomodulatory ability and their secretome production. Moreover, relevant to our delivery route for HSPCs, reduced blood flow typically occurs with increased age, which could impact the circulation and recruitment of systemically administered populations to the retina [20,21,22]. Notably, in the clinical context of dry AMD, these findings suggest that HSPCs from younger donors may lead to better therapeutic outcomes, a finding consistent with evidence from hematopoietic cell transplantation, where younger donors are associated with improved survival and transplant efficacy [27]. This aligns with broader regenerative medicine principles, where younger or rejuvenated progenitor cells often exhibit enhanced proliferative and differentiation potential [28]. The age of the animals used in this study could be perceived as a limitation. It is estimated that 1 human year is equivalent to 9 days of a mouse’s life [29].

Thus the “young” mice would equate to a 6-year-old human at the beginning of the experiment and a 19.5-year-old by the end, while the “old mice” would equate to a 50-year-old human at the beginning of the experiment and a 63-year-old by the end. Dry AMD has been reported in subjects as early as 50 years of age with an annual incidence of AMD increasing with age from <1% for those younger than 60 years to over 5% for people aged 80 years and above [30,31]. Thus, the age of our mice does fall within the age group in which dry AMD can be observed in humans. We did not use older animals due to the higher mortality seen in older mice which increases with age, which is significant from 18 months and over [32]. Importantly, we previously demonstrated that LV-RPE65 HSPC administration is most therapeutic in early disease [2], which further supports our timeline used in these studies. A potential concern for systemic administration is the LV-RPE65 HSPC potential to migrate to other injured tissues/organs besides the retina. We previously demonstrated that unincorporated HSPCs in normal mice will return to the bone marrow or spleen [33]. The systemic administration of GFP-labeled RPE65 HSPCs into the SOD2 KD mouse demonstrated that significant numbers of GFP-positive cells are observed in the retina, spleen, lung, and bone marrow at 1 day following injection [2]. The number of GFP^+^ cells decrease to non-significant levels in spleen, lung, and BM by 28 days post injection while a significant number of GFP^+^ cells remained in the retina at 3 months and beyond without any overt change in the health of the animals [2]. Moreover, allogeneic HS/PC have been administered to patients since 1957 for bone marrow transplantation; the cells enter the circulation and thus have access to all tissues and organs before they engraft in the bone marrow. Side effects are typically limited to histocompatibility [34].

Our approach has clear clinical potential for the treatment of AMD at early stages but requires functional human CD34^+^ cells, representing the progenitor population that is typically used for bone marrow transplantation [35] and is similar to the mouse ckit^+^Sca-1^+^ HSPCs used here and in our other studies [36]. Cord blood will likely not be available for the vast majority of aged individuals with AMD and banked CD34^+^ cells are not patient-specific and would require allogeneic cells to be obtained. Bone marrow mobilization or aspirate in the elderly could be potentially harmful to patients and will likely be found in aged donors, thus limiting the feasibility of using “fresh cells” in many patients. The use of induced pluripotent stem cells (iPSCs) derived from patients can be used to generate high numbers of CD34^+^ cells for genetic programing, and currently, there is a vast bank of iPSC-derived cells available for allogeneic use [37,38].

Cotrim and colleagues reported that the intravitreal use of a bone marrow mononuclear fraction containing CD34^+^ cells in a small cohort of patients with atrophic AMD was safe and resulted in a small improvement in best corrected visual acuity [39]. There are several advantages of hiPSC-CD34^+^ cells over those isolated from peripheral blood CD34^+^ cells: (1) iPSCs allow the generation of large numbers of CD34^+^ cells and the potential for long term cryogenic storage for immediate availability; (2) iPSC are considered “rejuvenated” and lose many of their aging features and epigenetic markers that are associated with disease [40,41]; (3) iPSC-derived cells have been shown to be less immunogenic than native phenotype [42,43]; and (4) iPSC allow the possibility of isogenic modification [44] to correct the genetic variants associated with AMD.

## 5. Conclusions

Our findings underscore the potential of systemic stem cell therapy with programmed HSPCs as a novel treatment strategy for AMD, particularly emphasizing the influence of donor and recipient age on therapeutic efficacy. These results pave the way for further investigation into age-related factors that may optimize stem cell therapies for AMD.

## Figures and Tables

**Figure 1 cells-14-01360-f001:**
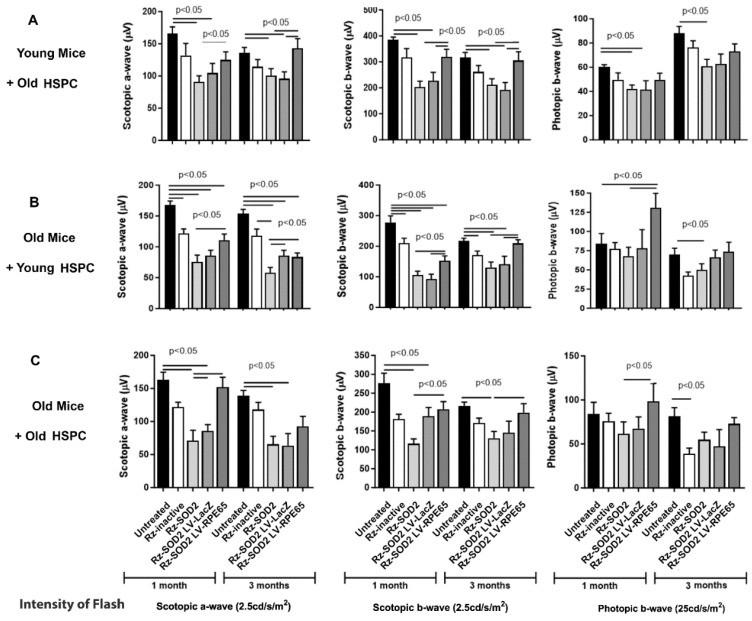
Age-dependent visual function recovery in SOD2 KD mice receiving LV-RPE65 HSPCs. Scotopic and photopic ERG analysis was performed at one and three months after systemic administration of reprogrammed young or old HSPCs to either young or old SOD2 KD mice [(**A**) old HSPCs → young mice; (**B**) young HSPCs → old mice; (**C**) old HSPCs → old mice]. The peak changes were quantified and demonstrated a significantly reduced ERG response in untreated SOD2 KD animals and SOD2 KD mice receiving LV-LacZ HSPC. Systemic administration of LV-RPE65 HSPC significantly reduced loss of scotopic a- and b-wave function in SOD2 KD mice irrespective of age of donor or host. No significant improvement was observed in the photopic b-wave at either 1 or 3 months following HSPC administration. The data (*n* ≥ 5 for each group) are presented as mean ± SEM and *p* < 0.05 was considered significant.

**Figure 2 cells-14-01360-f002:**
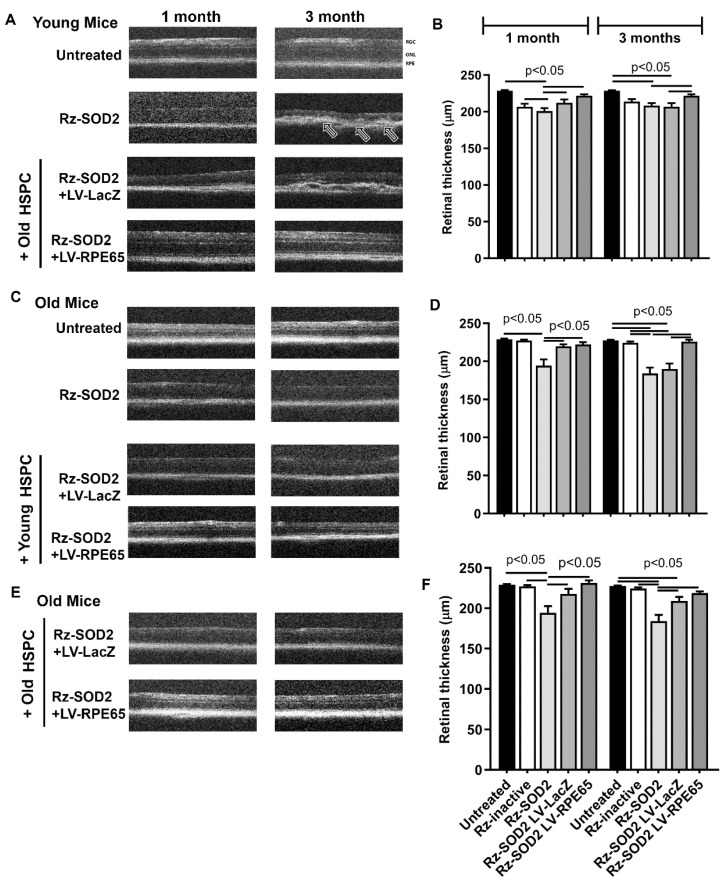
Optical coherence tomography from SOD2 KD mice receiving HSPCs. Representative optical coherence tomography images (arrows) and total retina thickness quantification after one and three months following systemic administration of reprogrammed young or old HSPCs to either young or old SOD2 KD mice [(**A**,**B**) old HSPC → young mice; (**C**,**D**) young HSPC → old mice; (**E**,**F**) old HSPC → old mice]. OCT analysis demonstrated that HSPC from either young or old mice maintained near normal retinal structure in both young and old SOD2 KD mice in contrast to SOD2 KD eyes alone which demonstrated severe abnormalities and retinal thinning. Quantification demonstrated that retinal thinning was significantly prevented at 1 and 3 months following LV-RPE65 HSPC treatment (*p* < 0.05). The data (*n* ≥ 5 for each group) are presented as mean ± SEM and *p* < 0.05 was considered significant.

**Figure 3 cells-14-01360-f003:**
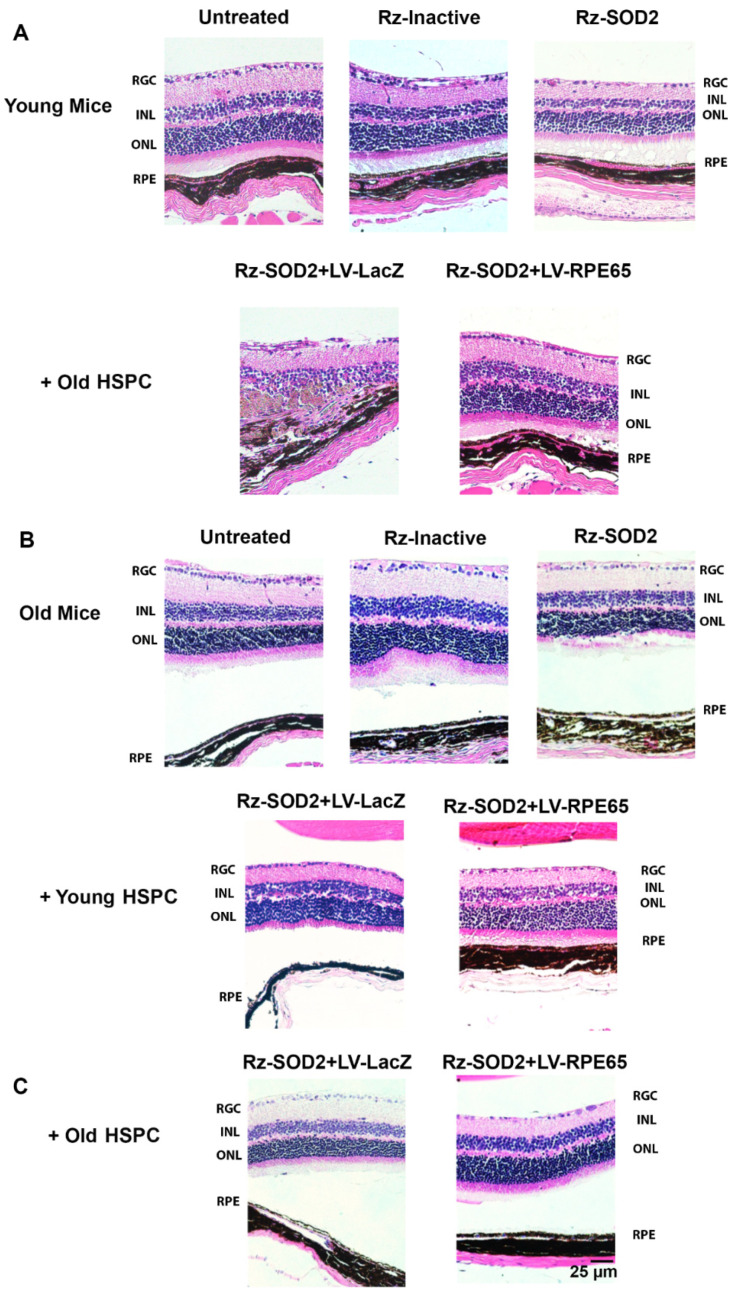
Systemically administered LV-RPE65 HSPCs reduce retinal pathology in the SOD2 KD mouse. Representative H&E-stained sections for three months following the systemic administration of reprogrammed young or old HSPCs to either young or old SOD2 KD mice [(**A**) old HSPC → young mice; (**B**) young HSPCs → old mice; (**C**) old HSPCs → old mice]. Retinal pathology, RPE changes and retinal thinning were observed in untreated SOD2 KD mice and SOD2 KD mice receiving LV-LacZ HSPCs. Neither old SOD2 KD mice receiving LV-RPE65 HSPCs from young mice nor young SOD2 KD mice receiving LV-RPE65 HSPCs from old mice presented with any major change in neural retina and RPE layer morphology (**A**,**B**), although in the case of young mice receiving old LV-RPE65 HPSCs, some cells were observed in the subretinal space overlying the RPE (**B**). LV-RPE65 HSPCs from old mice administered to old SOD2 KD mice showed less pathology compared to Rz-SOD2-only mice but there remained some neural retina thinning and RPE monolayer changes (**C**). *n* ≥ 3 for each group.

**Figure 4 cells-14-01360-f004:**
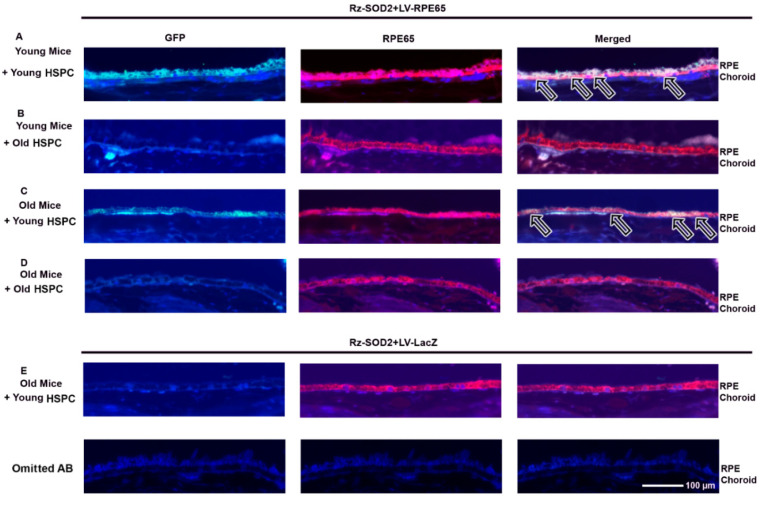
Immunostaining for RPE65 and GFP following the systemic administration of LV-RPE65 HSPC to SOD2 KD mice. Representative images are shown for three months following systemic administration of reprogrammed young and old HSPCs to either young or old SOD2 KD mice: (**B**) old HSPCs → young mice; (**C**,**E**) young HSPCs → old mice; (**D**) old HSPCs → old mice. Colocalization of GFP-positive staining (green) and RPE65 staining (red), indicative of integrated HSPCs (arrows), was observed at the RPE layer in young and old mice receiving HSPCs from young donors (**A**,**C**). Sections were counterstained with DAPI (blue). GFP-positive HSPCs were observed in the subretinal space of young mice receiving old HSPCs but not integrated into the RPE65-stained RPE monolayer (**B**). No GFP-positive cells were observed in the RPE layer or subretinal space of old SOD2 KD mice receiving old LV-RPE65 HSPCs (**D**) or in SOD2 KD mice receiving LV-LacZ HSPCs (**E**). No GFP or RPE65 staining was observed with omission of the primary antibody. *n* ≥ 3 for each group.

**Figure 5 cells-14-01360-f005:**
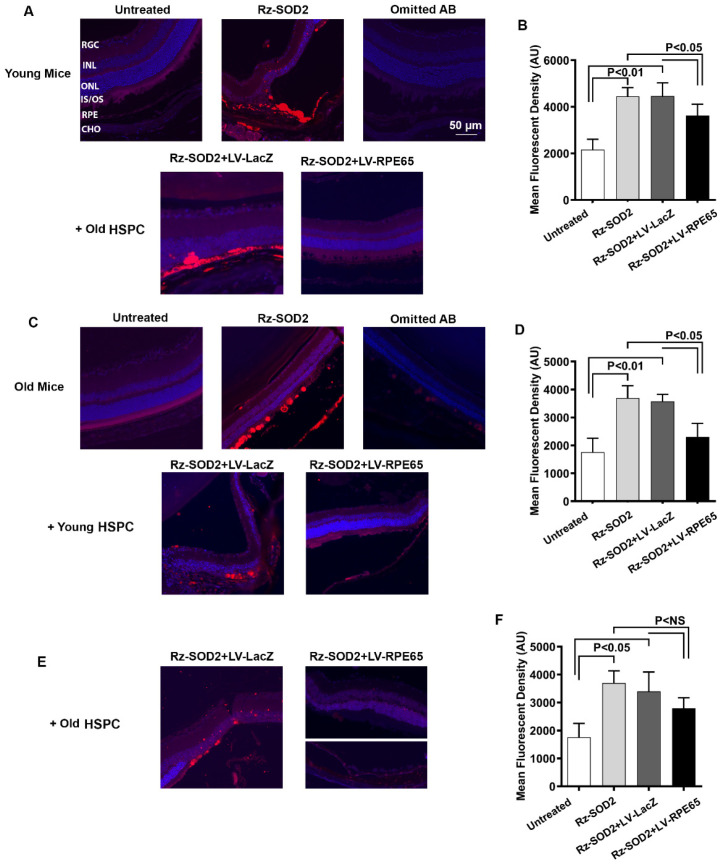
Systemically administered LV-RPE65 HSPCs reduce oxidative damage in SOD2 KD mice. Representative images and quantitation of fluorescence three months following systemic administration of reprogrammed young and old HSPCs to either young or old SOD2 KD mice: (**A**,**B**) old HSPCs → young mice; (**C**,**D**) young HSPCs → old mice; (**E**,**F**) old HSPCs → old mice. Retinas were immunostained for 4-hydroxynonenal (4-HNE, red)) and DAPI-stained (blue) to assess the degree of oxidative damage. SOD2 KD retinas demonstrated a large increase in 4-HNE immunostaining (red) relative to a very faint staining of 4-HNE in untreated retinas of both young and old mice. This increase in 4-HNE was not reduced by treatment with LV-LacZ HSPC (**A**–**D**) at *p* < 0.01 (mean ± SEM, *n* = 3). However, 4-HNE staining was significantly reduced in the retinas of young SOD2 KD mice receiving LV-RPE65 HSPCs from old donor mice and the retinas of old SOD2 KD mice receiving LV-RPE65 HSPCs from young donor mice (**A**–**D**) and this was statistically significant at *p* < 0.05 ((**B**,**D**); mean ± SEM, *n* = 3). We observed a small reduction in 4-HNE staining in old SOD2 KD mice retinas receiving LV-RPE65 HSPCs from old donor mice, but this was not significant when compared to untreated SOD2 KD mice or those receiving SOD2 KD + LacZ HSPCs (**E**,**F**) *p* < NS (mean ± SEM, *n* = 3).

**Figure 6 cells-14-01360-f006:**
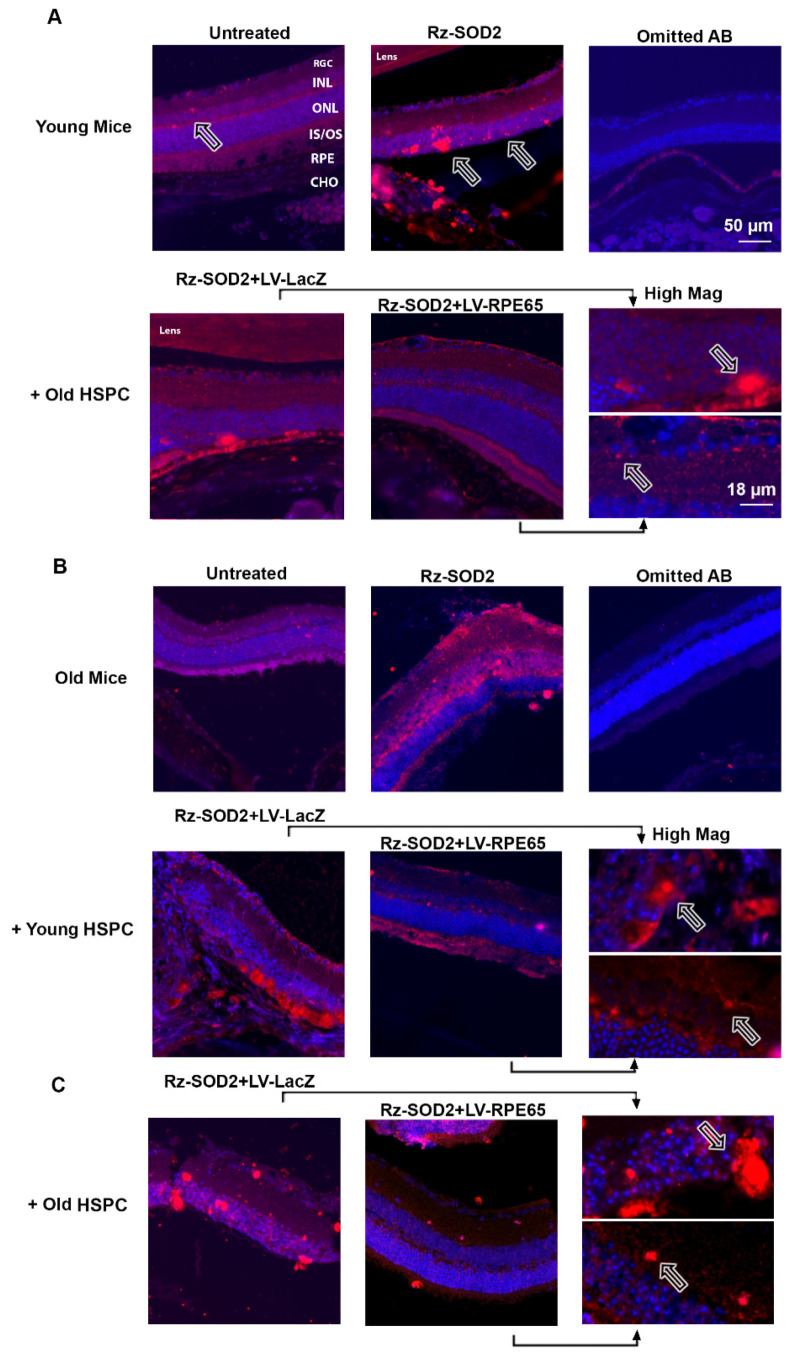
Systemically administered LV-RPE65 HSPCs reduce inflammation in SOD2-KD. Representative fluorescence images of mouse retinas immunostained for Iba1 (red; arrows) to identify microglia and DAPI-staining (blue): (**A**) old HSPCs → young mice; (**B**) young HSPCs → old mice; (**C**) old HSPCs → old mice. Untreated control retinas from young or old mice demonstrated very few small Iba1^+^ microglia that were scarcely distributed in the retina. By contrast, retinas from either young or old Rz-SOD2 KD mice demonstrated a large increase in large microglia in the retina and subretinal space (**A**,**B**). This increase in retinal microglia staining was not suppressed in SOD2 KD mice receiving LacZ-HSPCs but was greatly decreased in SOD2 KD mice receiving LV-RPE65 HSPCs from both young or old mice (**A**–**C**).

## Data Availability

The original contributions presented in this study are included in the article. Furtherinquiries can be directed to the corresponding author.

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
