# Peer review of "Impact of Donor and Host Age on Systemic Cell Therapy to Treat Age-Related Macular Degeneration"

_cells, 2025, doi:10.3390/cells14171360_

Round 1

Reviewer 1 Report

Comments and Suggestions for Authors

This is an interesting manuscript describing studies assessing the effects of age on cell therapy in AMD. In particular, the authors compared old and young programmed mouse hematopoietic bone marrow-derived progenitor cells (HSPC) effects on old and young SOD2KD (a model of AMD) mice. Functional and histological analyses determined that old cells and old mice demonstrated the lowest therapeutic efficacy. However, there was a general improvement that was most evident, as expected, in young mice treated with HSPC from young donors. Interestingly, the authors also observed an anti-inflammatory effect identified by analyzing microglia presence and morphology in the treated tissue. Overall, the bulk of the results provides some new information and warrants about the effect of age on cell-based therapy for AMD.

Few points should be clarified prior acceptance:

  1. Is there any difference in infection efficiency among old and young HSPC?
  2. In all the figures the word omitted is spelled “omited”, please correct.
  3. Ln 51 did you mean mediate?
  4. Ln 425 please add a reference.
  5. In the discussion it is not clear whether the authors are suggesting that the HSPC-based therapy is not a good alternative therapy whereas the iPSC-based is. Please, clarify.  

Author Response

Reviewer 1:

Few points should be clarified prior acceptance:

  1. Is there any difference in infection efficiency among old and young HSPC?

Response:  We did not observe any difference in infection efficiency between old and young HSPC.  This is now mentioned in the methods section.

  1. In all the figures the word omitted is spelled “omited”, please correct.

Response: Thank you for finding these errors and they have all been corrected. 

  1. Ln 51 did you mean mediate?

Response: This typo has been corrected.

  1. Ln 425 please add a reference.

Response: We appreciate you bringing this to our attention and now we have added the appropriate reference.

  1. In the discussion it is not clear whether the authors are suggesting that the HSPC-based therapy is not a good alternative therapy whereas the iPSC-based is. Please, clarify.  

Response: We have modified the discussion to make the point that HSPC using fresh cells from the patient or an allogeneic donor is ideal but in individuals where this is not possible the use of allogeneic iPSC differentiated into CD34+ cells is an excellent option (lines 357-364).  

Reviewer 2 Report

Comments and Suggestions for Authors

This manuscript describes work based on the project published by this lab in 2017 (ref 2,). The authors compared preprogrammed mouse hematopoietic bone marrow-derived progenitor cells (HSPC) from young or old donors and systemically transplanted them into young or old SOD2 KD mice. They showed that cells from young donors were better than from old donners, and young hosts are better than the old hosts for protective effects.

Of concern is the quality of the data presented. For example, the b-wave amplitude in 3-month untreated (presumably normal) young mice is significantly higher than that in 1-month untreated (90 vs 60 µV, Fig. 1A). This is quite different from Fig. 1D of the 2017 paper. Also, the photopic b-wave in old SOD2 KD mice treated with RPE65-HSPC is significantly higher than that of the untreated control (130 vs 80 µV, Fig. 1B, 1 month, p<0.05). This finding, if true, may suggest an increase in cone cells 1 month after RPE65-HSPC treatment. The authors may want to explain this finding. Flash intensities should be presented along with the scotopic a- and b-wave, and the photopic b-wave data, although 3 intensities for scotopic and 4 intensities for photopic ERG are listed in the Methods and Materials.

Several retinal sections presented in Fig. 3 had significant retinal detachment, making it difficult to see photoreceptor outer segments and the RPE. The representative section from a young SOD2 KD mouse (Fig. 3A, Rz-SOD2) had no outer nuclear layer (ONL) and thus all photoreceptors, including rods and cones were lost. However, the ERG (Fig. 1A) data indicate there were rod photoreceptors to generate scotopic a- and b-wave, and cones to generate photopic b-wave. OCT data (Fig. 2A) showed no missing ONL. In the 2017 paper, retinal sections in figure 5 of all had ONL.

It is interesting to see that retinas in the young SOD2 KD mice received old LacZ HSPC form rosettes (Figs. 2A (3-month), 5A, and 6A). This finding suggests that treatment with old LacZ HSPC induces retinal disorganization. The authors may want to discuss this finding.

Some text content could be more clearly written. For example, the two sentences in line 203-207 (started by “Previously”) linking the previous study to the present work is confusing. Also, the description of “untreated SOD2 KD mice” (line 216, 219) is not clear. In Fig. 1, a column was labeled as “untreated”, whereas another one as “Rz-SOD2”, but no “untreated SOD2 KD mice”. Retinal sections in Figs 2, 3, 5, and 6 should be labeled with retinal layers, including RPE, OS, ONL, etc.

Author Response

Reviewer 2

Of concern is the quality of the data presented. For example, the b-wave amplitude in 3-month untreated (presumably normal) young mice is significantly higher than that in 1-month untreated (90 vs 60 µV, Fig. 1A). This is quite different from Fig. 1D of the 2017 paper. Also, the photopic b-wave in old SOD2 KD mice treated with RPE65-HSPC is significantly higher than that of the untreated control (130 vs 80 µV, Fig. 1B, 1 month, p<0.05). This finding, if true, may suggest an increase in cone cells 1 month after RPE65-HSPC treatment. The authors may want to explain this finding. Flash intensities should be presented along with the scotopic a- and b-wave, and the photopic b-wave data, although 3 intensities for scotopic and 4 intensities for photopic ERG are listed in the Methods and Materials.

Response:  

We appreciate the reviewers’ careful assessment of both the 2017 paper and the current study. However, we are not surprised that the data presented in this study is not identical to that in the 2017 study since: 1) the 2017 study was undertaken almost a decade ago, in a different institution, using different equipment and by different personnel; and 2) as stated on the Jackson Labs web page “There is no such thing as a C57BL/6 mouse”  as there is significant genetic drift within C57BL/6 colonies requiring the need to repeatedly generate new substrains.  The substrain used in the 2017 paper was no longer have been available for the current study.

We appreciate the reviewers’ comment and have now acknowledged the possibility of an increase in cone cells 1 month after RPE65-HSPC treatment.  Flash intensities are now included in Figure 1.

Several retinal sections presented in Fig. 3 had significant retinal detachment, making it difficult to see photoreceptor outer segments and the RPE. The representative section from a young SOD2 KD mouse (Fig. 3A, Rz-SOD2) had no outer nuclear layer (ONL) and thus all photoreceptors, including rods and cones were lost. However, the ERG (Fig. 1A) data indicate there were rod photoreceptors to generate scotopic a- and b-wave, and cones to generate photopic b-wave. OCT data (Fig. 2A) showed no missing ONL.

Response:  We have improved the quality and replaced some of the representative images in Fig 3.  However, retinal detachment commonly occurs during processing of mouse eyes, especially those with some level of retinal pathology.

In the 2017 paper, retinal sections in figure 5 of all had ONL.

Response:  Please see the comment above.

It is interesting to see that retinas in the young SOD2 KD mice received old LacZ HSPC form rosettes (Figs. 2A (3-month), 5A, and 6A). This finding suggests that treatment with old LacZ HSPC induces retinal disorganization. The authors may want to discuss this finding.

Response: We respectfully disagree and do not see any rosettes in these images.   

Some text content could be more clearly written. For example, the two sentences in line 203-207 (started by “Previously”) linking the previous study to the present work is confusing.

Response: We have edited this section clarity and the revised section is now in lines 175-180.

Also, the description of “untreated SOD2 KD mice” (line 216, 219) is not clear.

Response: We apologize for any confusion but untreated SOD2KD mice have not received HSPC.

In Fig. 1, a column was labeled as “untreated”, whereas another one as “Rz-SOD2”, but no “untreated SOD2 KD mice”.

Response: We apologize if this is not clear but untreated is the WT mouse that has had no intervention whatsoever. The “Rz-SOD2” is the WT mouse that received the injection of the AAV-SOD2 ribozyme to generate the model. The “Rz-inactive” represent the WT mice that received AAV2 that express an inactive ribozyme.  

Retinal sections in Figs 2, 3, 5, and 6 should be labeled with retinal layers, including RPE, OS, ONL, etc.

Response: We appreciate the reviewer pointing this out to us and we have now corrected Figs 2-6 as requested.

Reviewer 3 Report

Comments and Suggestions for Authors

Title: Impact of donor and host age on systemic cell therapy to treat age-related macular degeneration

This manuscript addresses an important question regarding the impact of both donor and host age on the therapeutic efficacy of systemically delivered RPE65-programmed hematopoietic stem/progenitor cells (HSPCs) in an AMD-like murine model. The study is novel and has translational relevance for age-related macular degeneration (AMD) treatment. However, several points require clarification and improvement.

Major comments:

1, In Figure 1, the injection of Rz-inactive leads to a significant decrease in ERG responses compared to untreated controls. This is unexpected because the “inactive” ribozyme should not impact retinal function.

2, In some groups, injection of LV-LacZ HSPCs showed partial improvement in ERG and OCT compared to untreated animals, sometimes not significantly different from LV-RPE65-treated groups. This raises the question of whether the observed benefit is due to a non-specific effect of HSPC delivery.

3, According to H&E results, young mice injected with Rz-SOD2 develop more severe degeneration than old mice. This is counterintuitive, as older mice are typically more susceptible to oxidative stress.

4, Figure 4 shows that old donor HSPCs rarely integrate into the RPE layer. However, earlier results (ERG/OCT) show partial functional benefit in young hosts receiving old donor cells. How can this be explained? One hypothesis mentioned is early recruitment and transient survival of old cells. To strengthen this claim, additional early time-point experiments showing temporary RPE integration of old donor cells would be highly informative.

5, While quantitative analysis shows no significant decrease in 4-HNE levels with LV-LacZ treatment, the representative images suggest visible reduction in staining.

Minor comments:

1, Many figures do not display p-values on the graphs.

Author Response

Major comments:

1, In Figure 1, the injection of Rz-inactive leads to a significant decrease in ERG responses compared to untreated controls. This is unexpected because the “inactive” ribozyme should not impact retinal function.

Response: We agree with the reviewers’ comment that theoretically the expression of the inactive ribozyme should not have any effect; however, there are two potential issues. The inactive ribozyme represents a scrambled sequence that may have off target effects and impact on the retina and the process of performing a subretinal injection to administer the AAV-2 may cause some retinal damage.   

2, In some groups, injections of LV-LacZ HSPCs showed partial improvement in ERG and OCT compared to untreated animals, sometimes not significantly different from at one month treated groups. This raises the question of whether the observed benefit is due to a non-specific effect of HSPC delivery.

Response: We appreciate the reviewer bringing this to our attention and we have now addressed this in the discussion. For the ERG studies, the relevant comparison is between the “Rz-SOD2” and “Rz-SOD2LV-LacZ” and when these cohorts are examined the only time that the “Rz-SOD2LV-LacZ” improved was in the “old mice+young HSPS” cohort at 3 months (Scotopic a-wave) and Rz-SOD2” and “Rz-SOD2LV-LacZ”  in the “old mice+old HSPC” cohort at one month time point. This suggests that the rod photoreceptors, the main contributors to this ERG response, may preferentially benefit from HSPC administration. At one-month, retinal thickness similarly in all the experimental cohorts did show significant improvement in the mice receiving the LV-LacZ infected HSPC and at 3 months in the old mice receiving old HSPC. This supports that the HSPC independent of the age of the donor do have some reparative potential however, not to the same magnitude as the LV-RPE65 HSPC.      

3, According to H&E results, young mice injected with Rz-SOD2 develop more severe degeneration than old mice. This is counterintuitive, as older mice are typically more susceptible to oxidative stress.

Response: We thank the reviewer for bringing this important point to our attention and while it may be that the older mice are more susceptible to oxidative stress, they likely also have compensatory mechanisms in place to deal with the loss of SOD2 and may be better equipped than young mice. Age mice may have several compensatory mechanisms already activated to mitigate the effects of SOD2 deficiency. These would include upregulation of other antioxidant systems such as SOD1 (cytosolic Cu/Zn-SOD) and SOD3 (extracellular SOD) to partially compensate for the loss of SOD2. Also, catalase and glutathione peroxidase (GPx) may be activated to detoxify hydrogen peroxide (Hâ‚‚Oâ‚‚), the product of superoxide dismutation, helping to maintain redox homeostasis. Also, the activation of redox-sensitive transcription factors such as NRF2 could be activated to induce expression of detoxifying and antioxidant genes. Cells of the aged retina may undergo “metabolic reprogramming” and shift metabolism to reduce mitochondrial ROS production, such as increasing reliance on glycolysis over oxidative phosphorylation and enhancing mitophagy to remove damaged mitochondria.

4, Figure 4 shows that old donor HSPCs rarely integrate into the RPE layer. However, earlier results (ERG/OCT) show partial functional benefit in young hosts receiving old donor cells. How can this be explained?

Response: We apologize for not making this point clear, but even old HSPC are still capable of providing paracrine factors that support the survival of the RPE cells. Integration is not necessary, just the presence of HSPC within the vicinity to send survival factors to the injured RPE.  

One hypothesis mentioned is early recruitment and transient survival of old cells. To strengthen this claim, additional early time-point experiments showing temporary RPE integration of old donor cells would be highly informative.

Response: We appreciate this comment and agree with the reviewer, but funding is no longer available for additional studies. 

5, While quantitative analysis shows no significant decrease in 4-HNE levels with LV-LacZ treatment, the representative images suggest visible reduction in staining.

Response: We are presenting representative images from each group and as shown in the quantitation there is some variation between mice in each group.  However, we have re-examined the images presented in this paper and are unable to conclude any visible reduction in staining in 4-HNE levels with LV-LacZ treatment.

Minor comments:

1, Many figures do not display p-values on the graphs.

Response: This has now been addressed

Reviewer 4 Report

Comments and Suggestions for Authors

The work stands out due to its high substantive quality as well as its clear and logical structure. The topic is presented in a thoughtful and thorough manner, demonstrating a solid understanding of the relevant literature and current research trends. The methodology applied is appropriate to the stated objectives, and the results have been carefully analyzed and clearly presented.

Despite its strong substantive value, the article requires some minor editorial and visual adjustments. I would like to point out the following issues:

  1. Repetitions – The reference to Figure 1 appears multiple times; I suggest keeping it only in line 209, where it is most appropriate and logically placed.
  2. Unnecessary blank lines – In some sections, unjustified breaks or empty lines disrupt the flow of reading. It would be advisable to remove them to ensure the text is more concise.
  3. Abbreviations – Numerous abbreviations are used throughout the article, some of which are not explained upon first mention. I recommend introducing them consistently and considering the inclusion of an abbreviations list at the end of the article, which would help readers—particularly those not specialized in the field—navigate the content more easily.

A final editorial revision is recommended prior to publication.

Author Response

Reviewer 4

Despite its strong substantive value, the article requires some minor editorial and visual adjustments. I would like to point out the following issues:

  1. Repetitions – The reference to Figure 1 appears multiple times; I suggest keeping it only in line 209, where it is most appropriate and logically placed.

Response: We have removed the duplicate reference to Figure 1 as requested. 

  1. Unnecessary blank lines – In some sections, unjustified breaks or empty lines disrupt the flow of reading. It would be advisable to remove them to ensure the text is more concise.

Response: The empty lines have been removed as requested.

  1. Abbreviations – Numerous abbreviations are used throughout the article, some of which are not explained upon first mention. I recommend introducing them consistently and considering the inclusion of an abbreviations list at the end of the article, which would help readers—particularly those not specialized in the field—navigate the content more easily.

Response: We thank the reviewer for this comment and have identified all      abbreviations and corrected their first use.

Round 2

Reviewer 1 Report

Comments and Suggestions for Authors

The authors have responded to my previous comments. I have no further recommendations.

Author Response

We thank the reviewer for providing constructive comments though out the review process.  

Reviewer 2 Report

Comments and Suggestions for Authors

No significant improvement was found in the revised manuscript.

In response to the increase in the photopic b-wave in old SOD2 KD mice treated with RPE65-HSPC, the authors acknowledged and wrote that “in old SOD2 KD mice treated with RPE65-HSPC is significantly higher than that of the untreated control may suggest an increase in cone cells 1 month after treatment” (line 379-382).

It is common knowledge that mammals lack the ability to regenerate photoreceptors after the retina matures. This reviewer does not believe mice treated with RPE65-HSPC could lead to a significant increase in cone cells in 15 months old mice and believes it could be an experimental error. If the finding were true, it could be a monumental discovery and could lead to potential therapy for retinal degenerative diseases.

The authors claimed that they could not find any retinal rosette in their images. Retinal rosettes are formed by cells arranged in a circular, rosette-like pattern. They primarily consist of cells from the outer nuclear layer. Retinal rosettes can be associated with retinal dysplasia or with degenerative processes following retinal injury. It is rather easy to recognize retinal rosettes in Fig. 5A and 6A (green arrows).

The images with green arrows are in the attached PDF file.

Author Response

No significant improvement was found in the revised manuscript.

Response: We have modified the second revision to include new images of panels in Figures 5 & 6.   

In response to the increase in the photopic b-wave in old SOD2 KD mice treated with RPE65-HSPC, the authors acknowledged and wrote that “in old SOD2 KD mice treated with RPE65-HSPC is significantly higher than that of the untreated control may suggest an increase in cone cells 1 month after treatment” (line 379-382).

It is common knowledge that mammals lack the ability to regenerate photoreceptors after the retina matures. This reviewer does not believe mice treated with RPE65-HSPC could lead to a significant increase in cone cells in 15 months old mice and believes it could be an experimental error. If the finding were true, it could be a monumental discovery and could lead to potential therapy for retinal degenerative diseases.

Response: We apologize for not phrasing the sentence correctly. We intended to say there was improved function of cone driven ON bipolar cells based on improvement in the ERGs photopic b-wave. We have corrected this in the revised text.

The authors claimed that they could not find any retinal rosette in their images. Retinal rosettes are formed by cells arranged in a circular, rosette-like pattern. They primarily consist of cells from the outer nuclear layer. Retinal rosettes can be associated with retinal dysplasia or with degenerative processes following retinal injury. It is rather easy to recognize retinal rosettes in Fig. 5A and 6A (green arrows).

Response: We appreciate the reviewers  comments and have provided new images for those in question that are without any evidence of rosettes or folding of the retina in Figure 5 &6.   

Reviewer 3 Report

Comments and Suggestions for Authors

I have carefully reviewed the revised manuscript. I have no further comments.

Author Response

We thank the reviewer for the helpful comments through out this review process.